# A localization transition underlies
# the mode-coupling crossover of glasses

**Daniele Coslovich**[1*]**, Andrea Ninarello**[1,2]**, Ludovic Berthier**[1]

**1** Laboratoire Charles Coulomb (L2C), Université de Montpellier, CNRS, Montpellier, France
**2** CNR-ISC, UOS Sapienza, Piazzale A. Moro 2, 00185 Roma, Italy

⋆ daniele.coslovich umontpellier.fr

## Abstract

We study the equilibrium statistical properties of the potential energy landscape of several glass models in a temperature regime so far inaccessible to computer simulations. We show that unstable modes of the stationary points undergo a localization transition in real space close to the mode-coupling crossover temperature determined from the dynamics. The concentration of localized unstable modes found at low temperature is a non-universal, finite dimensional feature not captured by mean-field glass theory. Our analysis reconciles, and considerably expands, previous conflicting numerical results and provides a characteristic temperature for glassy dynamics that unambiguously locates the mode-coupling crossover.



# 1   Introduction

The formation of a glass from the supercooled melt results from a giant increase of the structural relaxation time when the temperature drops below the melting point [1, 2]. Whether the slowing down of molecular motion is driven by a single or several physical mechanisms, active over distinct temperature regimes, is still unclear. Available theories are all but unanimous [3, 4], while experiments and simulations, despite recent technical and numerical advances [5], struggle to disentangle theoretical predictions on the sole basis of relaxation data. The existence of a temperature crossover separating two physical regimes of dynamic relaxation is supported by a number of empirical observations and models, but is subject to lively debates [4]. This crossover is described as an avoided dynamic singularity by mode-coupling [6] and mean-field [7] theories of glasses.

In the early 2000's, a series of studies [8–13] suggested that this smooth dynamic crossover originates from an underlying sharp geometric transition characterizing the potential energy surface (PES). Physically, the transition is between a high-temperature regime where the dynamics takes place near unstable saddle modes and a low-temperature one where dynamics is activated between energy minima. In mean-field glass models, a geometric transition is analytically found: below a critical energy level, the closest stationary point to a typical configuration on the PES is not a saddle anymore but a local minimum [14, 15]. The existence of a geometric transition was reported for several liquids with soft interactions, with universal characteristics [9–12, 16]. However, these results have been criticized at the conceptual [17] and methodological levels [18, 19]. In particular, the early studies on the geometric transition employed an optimization method that locates stationary points only rarely, the vast majority of optimized configurations being "quasi-stationary" points characterized by small, non-vanishing force and precisely one inflection mode [12, 19]. Subsequent studies did not find a transition [20–23], but the transition temperature could not be easily crossed at equilibrium. From these conflicting results it is difficult to draw firm conclusions on the nature of the mode-coupling crossover in actual three-dimensional liquids.

Here, we resolve these contradictions and clarify the nature of the change of the PES associated to the mode-coupling crossover in finite dimensions. Our work builds on two key enabling factors, respectively algorithmic and conceptual. The first key feature of our work is the use an efficient swap Monte Carlo algorithm [24, 25], which enables us to probe the landscape properties on both sides of the mode-coupling crossover temperature [26, 27]. Second, we recognize that the geometric transition, as obtained in mean-field models, can only concern the subset of unstable directions on the PES that correspond to *delocalized* displacements, which involve a finite fraction of particles. Previous studies of the statistics of stationary points have considered instead all unstable modes, irrespective of their spatial characteristics. Our analysis demonstrates that a geometric transition occurs only for delocalized modes and that the mode-coupling crossover therefore coincides with a localization transition of the unstable directions of the PES. We argue that the extent to which the mode-coupling singularity is avoided in real liquids is controlled by the concentration of *localized* modes (involving a finite number of particles), which is found to be system-dependent. Finally, we pinpoint one qualitative difference between the features of stationary and quasi-stationary points, but also confirm that the statistical properties of these two sets of data yield identical results for the localization transition identified in the present work.

## 2 Methods

We determined stationary and quasi-stationary points of the potential energy surface (PES) for systems of $N$ point particles using two different optimization methods. The first method consists in straightforward minimization of the total squared force

$$W = \frac{1}{N} \sum_{i=1}^{N} |\vec{F}_i|^2, \tag{1}$$

where $\vec{F}_i$ is the force on particle $i$. Minimizations start from instantaneous configurations obtained from Monte Carlo or molecular dynamics simulations at a given number density $\rho = N/V$ and temperature $T$. For each configuration, we used the l-BFGS minimization algorithm [28] to minimize $W$. It is well-known that $W$-minimizations locate true stationary points only rarely [29] and that the vast majority of points determined with this method are quasi-stationary points, at which there is precisely one inflection mode having a null zero eigenvalue [19]. In our minimizations, this inflection mode has a nearly zero eigenvalue whose norm $|\lambda|$ is typically lower than $10^{-4}$ (in the corresponding reduced units, see below) and which is clearly distinguishable from the lowest non-zero eigenvalue for the system sizes used in this work. The inflection mode was removed from the analysis, to avoid spurious $O(1/N)$ finite size effects when the fraction of unstable modes gets close to zero.

The stationary and quasi-stationary points obtained from $W$-minimizations can be distinguished on the basis of the corresponding value of $W$ (in reduced units), which is low but non-zero for quasi-stationary points and zero within machine precision for true stationary points ($W \sim 10^{-14}$). In practice, we use a threshold of $\sim 10^{-10}$ to classify the two kinds of points for all models except for the polydisperse spheres with $n = 12$ (see below), for which a slightly higher threshold is used ($3 \times 10^{-9}$) to account for a less strict convergence criterion on $W$-minimizations. Points that were not recognized as either stationary points or quasi-stationary points according to the above criteria were removed from the analysis. Previous studies showed that the statistical properties of quasi-stationary points and stationary points are practically indistinguishable above $T_{\mathrm{MCT}}$ [29]. We discuss the similarity and differences between these two kinds of points further down in the manuscript and in the Appendix.

To corroborate our analysis, we performed additional optimizations using the eigenvector-following (EF) method introduced by Wales [30]. The EF method is a generalization of the Newton-Rapshon method and locates stationary points containing a prescribed number of unstable modes $n_u$. At each iteration, the Hessian matrix is diagonalized yielding a set of (local) $3N$ normal modes with eigenvalue $\lambda_\alpha$ and (normalized) eigenvector $\vec{e}_\alpha$. The elementary step along mode $\alpha$ is defined as

$$\Delta x_\alpha = S_\alpha \frac{2G_\alpha}{|\lambda_\alpha|(1 + \sqrt{1 + (4G_\alpha/\lambda_\alpha)})}, \tag{2}$$

where $G_\alpha = \vec{G} \cdot \vec{e}_\alpha$ is the projection of the gradient $\vec{G}$ on the mode and the signs $S_\alpha = \pm 1$ are defined below. At each iteration, the particles' positions are displaced by $\Delta \vec{x} = K \sum_\alpha \Delta x_\alpha \vec{e}_\alpha$, where $K$ is a scaling factor that ensures that the amplitude of each displacement $K \Delta x_\alpha$ lies within a mode-dependent trust radius $\delta_\alpha$ [30]. The trust radii are initially set to $\delta_\alpha = 0.2$ and are increased (decreased) at each iteration by a factor 1.2 if the relative error $r = (\lambda_{est} - \lambda_\alpha)/\lambda_\alpha$ is smaller (or larger) than 1. The trust radius can vary in the interval $10^{-7} < \delta_\alpha < 1$. $\lambda_{est}$ is a first-order approximation to the current eigenvalue $\lambda_\alpha$, for which we used an improved expression that accounts for changes in the (local) eigenvectors, as given by Ruscher [31].

In the EF method, the signs $S_\alpha$ are fixed at the beginning of the optimization. If $S_\alpha = -1$ (+1), the algorithm searches for a minimum (maximum) along mode $\alpha$. The number of negative signs thus defines the order $n_u$ of the searched stationary point. In subsequent studies

of the PES of glass-forming liquids [20, 32], the target value of $n_u$ was not fixed from the outset and a Netwon-Raphson-like step was used instead, setting $S_\alpha$ to the sign of the (local) eigenvalue $\lambda_\alpha$. This hybrid scheme displayed convergence issues and therefore, in practice, the values of $S_\alpha$ were fixed to the ones found after $M = 20$ iterations. The order of the target stationary point thus depends indirectly on the initial trust radii and on the choice of the parameter $M$: higher values of $M$ lead to smaller $n_u$, but also decrease the success rate, *i.e.* the fraction of converged optimizations.

In a first implementation, we followed the hybrid scheme of Ref. [20] and confirmed that the choice $M = \infty$ leads to convergence issues, even at low temperature, in agreement with a recent study by Ruscher [31]. Our convergence criterion required that $W$ drops below $10^{-10}$ within a maximum of 4000 iterations. We note that the problem is not specific to the complex PES of glass-forming liquids and can be observed under some circumstances even in the simple case of the Müller-Brown surface. It can be tracked down to the fact that when a mode goes smoothly through a zero curvature region, the corresponding elementary step becomes dominant in magnitude and reverts its sign. When this occurs, the optimization oscillates back and forth along a soft direction and the overall behavior of the algorithm becomes erratic.

To overcome these difficulties and to avoid that our results depend on choice of $M$, we decided to fix the number of unstable modes from the outset. For each configuration, we first determined the number of unstable modes found after the corresponding $W$-minimization and then target this same value during the EF optimization. Although our approach does not provide an independent mapping between instantaneous configurations and stationary points, it is more robust than the hybrid EF method and enables in addition a useful and straightforward comparison between the properties of stationary points and the more general points located by $W$-minimizations.

# 3 Models

## 3.1 50-50 soft spheres

This is the historical 50:50 binary mixture introduced by Bernu *et al.* [33]. The pair interaction potential is

$$u_{\alpha\beta}(r) = \epsilon \left( \frac{\sigma_{\alpha\beta}}{r} \right)^{12}, \tag{3}$$

where $\alpha, \beta = A, B$ are species indexes. The size ratio is $\frac{\sigma_{AA}}{\sigma_{BB}} = 1.2$ and the cross-interaction term is additive $\sigma_{AB} = (\sigma_{AA} + \sigma_{BB})/2$. The potential is cut off and shifted at a distance $r_{cut} = \sqrt{3}\sigma_{AA}$ by adding a cubic term that ensures continuity of the potential up to the second derivative at $r_{cut}$ [11, 25]. Energies and distances are expressed in units of $\epsilon$ and $\sigma_{AA}$, respectively. We used configurations from previous molecular dynamics simulations for $N$ particles at a number density $\rho = N/V = 1$, with $N = 400, 800, 2000$ [34].

## 3.2 Ternary mixture

The ternary mixture model studied in this work was introduced by Gutierrez *et al.* in Ref. [35]. The interaction potential is given by inverse power laws with an exponent 12, plus additional terms that ensure continuity of the derivatives at the cutoff:

$$u_{\alpha\beta}(r) = \left( \frac{\sigma_{\alpha\beta}}{r} \right)^{12} + c_4 \left( \frac{\sigma_{\alpha\beta}}{r} \right)^{-4} + c_2 \left( \frac{\sigma_{\alpha\beta}}{r} \right)^{-2} + c_0, \tag{4}$$

where $\alpha, \beta = A, B, C$. The expressions for $c_0$, $c_2$, and $c_4$ are given in [36]. The size ratio between two species is $\frac{\sigma_{AA}}{\sigma_{BB}} = \frac{\sigma_{BB}}{\sigma_{CC}} = 1.25$, with additive cross-interactions, and the chemical

compositions are $x_A = 0.55$, $x_B = 0.30$, and $x_c = 0.15$. The potential is cut off at a distance $r_{cut} = 1.25\sigma_{\alpha\beta}$. We performed swap Monte Carlo simulations for $N = 250, 500, 1500, 3000$ particles at a number density $\rho = 1.1$. We used 80% of displacement moves over cubes of side $0.14\sigma_{AA}$ and 20% of swap moves [27]. To save computational time, we never attempted to exchange the identity of species $A$ and $C$. We note that this model liquid can be equilibrated with swap Monte Carlo below the mode-coupling temperature $T_{MCT} = 0.29$ [27]. However, because of its crystallization tendency at low temperature, we could not simulate the metastable liquid with $N = 1500$ particles for $T < 0.27$ and the one with $N = 3000$ particles for $T < 0.28$. Energies and distances are expressed in units of $\epsilon$ and $\sigma_{AA}$, respectively.

## 3.3 Network liquid

The network liquid model is a simple binary mixture that mimics the structure and dynamics of silica [37]. The interaction potential between unlike species ($\alpha \neq \beta$) is of the Lennard-Jones type

$$u_{\alpha\beta}(r) = 4\epsilon_{\alpha\beta}\left[\left(\frac{\sigma_{\alpha\beta}}{r}\right)^{12} - \left(\frac{\sigma_{\alpha\beta}}{r}\right)^{6}\right], \tag{5}$$

while the one between equal species is a simple inverse power

$$u_{\alpha\alpha} = \epsilon_{12}(\sigma/r)^{12}. \tag{6}$$

Energies and distances are expressed in units of $\epsilon_{AA}$ and $\sigma_{AA}$, respectively. The remaining interaction parameters are $\epsilon_{AB} = 6$, $\epsilon_{BB} = 1$, $\sigma_{AB} = 0.49$, $\sigma_{BB} = 0.85$. The potential is cut off smoothly at $r_{cut}$ by adding a cubic term that ensures continuity of the second derivative at the cut-off distance $r_{cut}$, as for the soft sphere mixture [11]. The resulting cut-off distances are 2.07692, 1.39081, 1.76538 for $A-A$, $A-B$ and $B-B$ interactions, respectively. We analyzed simulations for system sizes $N = 400, 800, 2000$ at a number density $\rho = 1.655$ obtained from previous molecular dynamics simulations [34].

## 3.4 Polydisperse particles n=18

We consider the model of polydisperse repulsive particles with additive interactions studied in Ref. [27]. The interaction potential between particles $i$ and $j$ is

$$u(r_{ij}) = \epsilon(\sigma_{ij}/r_{ij})^{n} + c_4\left(\frac{r_{ij}}{\sigma_{ij}}\right)^{4} + c_2\left(\frac{r_{ij}}{\sigma_{ij}}\right)^{2} + c_0, \tag{7}$$

with $n = 18$ and $\sigma_{ij} = (\sigma_i + \sigma_j)/2$. The coefficients $c_0$, $c_2$, $c_4$ are determined to ensure continuity of the potential at the cut-off distance $r_{cut} = 1.25\sigma_{ij}$, as for the ternary mixture. The distribution of particle diameters is $P(\sigma) = A/\sigma^3$ for $\sigma_{max} \leq \sigma \leq \sigma_{min}$ and 0 otherwise, with $A$ a normalization constant. We use $\sigma_{max}/\sigma_{min} = 2.219$, which implies a root mean square deviation of the diameter

$$\delta = \frac{\sqrt{\langle\sigma^2\rangle - \langle\sigma\rangle^2}}{\langle\sigma\rangle} \tag{8}$$

of about 23%. We simulated systems composed of $N = 500, 1000, 1500$ particles at a number density $\rho = 1$ using the swap Monte Carlo algorithm described in Ref. [27].

## 3.5 Polydisperse particles n=12

This is a variant of the polydisperse mixture introduced in the previous section. It features non-additive interactions to stabilize the fluid against phase separation [27]. The interaction potential between particles $i$ and $j$ is

$$u(r_{ij}) = \epsilon(\sigma_{ij}/r_{ij})^n + c_4\left(\frac{r_{ij}}{\sigma_{ij}}\right)^4 + c_2\left(\frac{r_{ij}}{\sigma_{ij}}\right)^2 + c_0, \tag{9}$$

with $n = 12$ and $\sigma_{ij} = (1-0.2|\sigma_i-\sigma_j|)(\sigma_i+\sigma_j)/2$. The coefficients $c_0$, $c_2$, $c_4$ are determined to ensure continuity of the potential at the cut-off distance $r_{cut} = 1.25\sigma_{ij}$. We use the same size distribution as for the polydisperse particles with $n = 18$. We simulated systems composed of $N = 500, 1000, 1500$ particles at a number density $\rho = 1$ using the swap Monte Carlo algorithm described in Ref. [27].

## 4 Results

We first present our central result and then provide its numerical support. Further details are given in the Supplementary Information (SI) of the accompanying dataset [38]. In Fig. 1 we show results of $W$-minimizations that locate stationary and quasi-stationary points of the PES (the distinction between these two families [19] does not affect our conclusions, see Appendix). We also include results for stationary points obtained using the EF method for the ternary mixture model. Diagonalization of the Hessian matrix yields a set of $3N$ eigenmodes, of which $n_u$ corresponds to negative eigenvalues $\lambda$, i.e., to unstable directions. The left panel shows the corresponding average fraction of unstable modes $f_u = n_u/(3N)$ as a function of temperature $T$ for all the models of glass-forming liquids we investigated. Most of these models have been equilibrated below their respective MCT crossover temperatures $T_{MCT}$, as determined from power-law fits of the relaxation time data [27, 37]. We are thus in a position to probe the landscape properties in a temperature range inaccessible to previous studies. At low temperature, we observe marked system-dependent deviations from an empirical power-law singularity introduced in Ref. [16]. Most importantly, the fraction of unstable modes is insensitive to the crossover and remains finite far below $T_{MCT}$, with barely any system-size dependence (see the SI [38]).

The picture changes qualitatively when the spatial localization of the modes is taken into account. As described below, we distinguish between localized and delocalized unstable modes via a finite size scaling analysis of the participation ratio. The fraction of delocalized unstable modes, $f_{ud}$, goes strictly to zero for all investigated fragile liquids at a temperature close to

Table 1: Mode-coupling temperatures $T_{MCT}$, localization transition temperatures $T_\lambda$, fitting parameter $A_\lambda$ and threshold energies $e_{th}$ for all the studied models. Note that for the network liquid no localization transition could be found.

| | $T_{MCT}$ | $T_\lambda$ | $A_\lambda$ | $e_{th}$ |
|---|---|---|---|---|
| 50-50 soft spheres | 0.20 [34] | 0.197 ± 0.002 | 11.7 ± 0.5 | 1.735 ± 0.015 |
| Ternary mixture | 0.288 [27] | 0.280 ± 0.002 | 18.0 ± 1.1 | 1.077 ± 0.012 |
| Ternary mixture (EF) | | 0.287 ± 0.010 | 19 ± 7 | 1.10 ± 0.01 |
| Network liquid | 0.31 [34] | – | – | – |
| Polydisperse particles $n = 18$ | 0.50 [27] | 0.460 ± 0.003 | 20.0 ± 1.0 | 1.47 ± 0.02 |
| Polydisperse particles $n = 12$ | 0.104 [27] | 0.086 ± 0.004 | 3.4 ± 0.4 | 0.22 ± 0.01 |

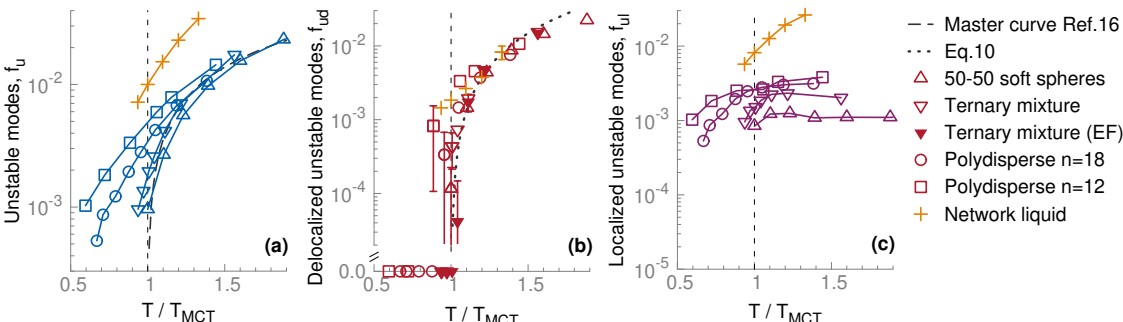

Figure 1: Fractions of unstable modes in stationary and quasi-stationary points for all studied models. Rescaled temperature dependence of (a) total fraction $f_u$ of unstable modes from $W$-minimizations. The dashed line is the master curve found in Ref. [16]. (b) Fraction $f_{ud}$ of delocalized unstable modes. Open and filled symbols correspond to $W$-minimizations an EF optimizations, respectively. The dotted line is Eq. (10) with $f_0 = 0.042$. Error bars are shown when the relative error exceeds 20%. (c) Fraction of localized unstable modes from $W$-minimizations.

$T_{\text{MCT}}$, see Fig. 1(b). The approach to $T_{\text{MCT}}$ can be described approximately by the following power law

$$f_{ud} = f_0 \left( T / T_{\text{MCT}} - 1 \right)^{3/2}, \quad \text{if } T > T_{\text{MCT}}. \tag{10}$$

The exponent 3/2 in Eq. (10) describes the approach to the dynamical transition in the mean-field $p$-spin model [39]. As in mean-field, a geometric transition occurs indeed at a finite temperature but it captures the disappearance of delocalized unstable modes only. In finite dimensions, localized modes exist at any temperature and therefore the MCT crossover coincides with a localization transition of unstable modes. We emphasize that the values of $T_{\text{MCT}}$ were independently estimated from power law fits to the relaxation time data elsewhere, see Table 1, and need not coincide exactly with the temperature at which $f_{ud}$ vanishes. In particular, we observe some discrepancy for polydisperse particles, which likely reflect the uncertainty inherent in the fits to the dynamic data, *e.g.* the choice of the temperature range. The statistical uncertainty on $f_{ud}$ becomes large, in relative terms, only when approaching the transition from the right. Close inspection of the data in proximity to the transition (see also Fig. 11 in the Appendix) indicates that the fraction of delocalized modes decreases faster in stationary points than in quasi-stationary points. This feature can be tracked down to the different shape of the corresponding spectra at low temperature, see Fig. 6 below. Nonetheless, both sets of data consistently vanish below the estimated mode-coupling temperature.

Figure 1(c) shows that the concentration of localized unstable modes $f_{ul} = (f_u - f_{ud})$ is system-dependent. We suggest that a higher fraction of localized modes at $T_{\text{MCT}}$ corresponds to stronger deviations from the geometric transition scenario observed in the mean-field $p$-spin model. Liquids that narrowly avoid the geometric transition should display a marked change in dynamic behavior across the MCT crossover temperature. On the other hand, we note that there exist models of glassy dynamics whose PES is, unlike the one of $p$-spins, trivial or not smooth and yet display MCT-like dynamics [40–42]. Whether another mechanism can explain the MCT phenomenology in all these systems remains a challenging open question. We expect hard spheres to be similarly characterized by a localization transition of unstable modes in a free-energy landscape, but there exist at present no computational tool to attack this problem. Finally, the trend in Fig. 1(c) superficially suggests a correlation between glass-forming ability [27] and the concentration of localized unstable modes. We observe that polydisperse particles have the largest concentrations of localized modes and are found to be very stable against crystallization even when using efficient swap Monte Carlo techniques [27], while the

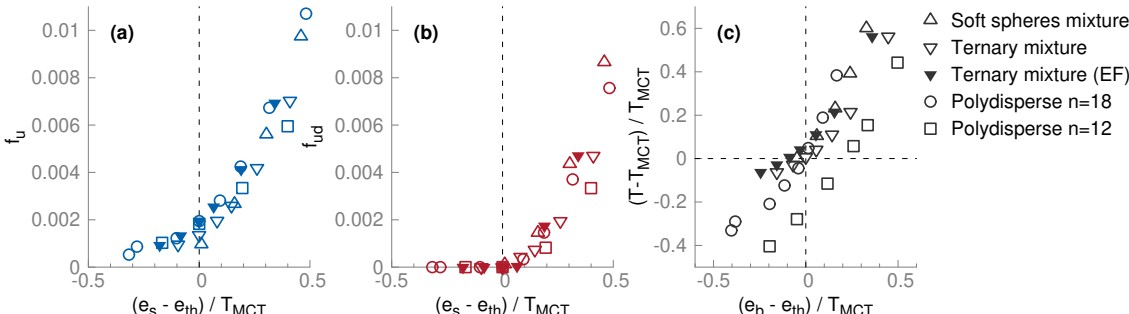

Figure 2: Test of the geometric transition scenario. (a) Fraction of unstable modes $f_u$ and (b) fraction of delocalized unstable modes $f_{ud}$ as a function of the scaled energy $(e_s - e_{th})/T_{\mathrm{MCT}}$ for points obtained from $W$-minimizations (open symbols) and EF optimizations (filled symbols). Temperature is used as an implicit variable. (c) Scaled temperature $(T - T_{\mathrm{MCT}})/T_{\mathrm{MCT}}$ versus scaled bare energy $(e_b - e_{th})/T_{\mathrm{MCT}}$.

50-50 soft sphere mixture has the lowest concentration of localized modes and can be crystallized with conventional simulation methods. While intriguing, this observation needs to be tested over a more diverse pool of liquids and also changing the dimensionality of space.

In mean-field $p$-spin models, the geometric transition is defined through the relationship between $f_u$ and the energy $e_s$ of stationary points. At the transition, stationary points of average energy $e_{th}$ have a vanishing fraction of unstable modes $f_u$: $f_u(e_{th}) = 0$ [11]. Such a representation reveals an intrinsic property of the landscape and it is fairly insensitive to the way in which the latter is sampled. In finite-dimensional systems, the $e_s(f_u)$ relation deviates from the linear behavior observed well above $T_{\mathrm{MCT}}$ [22,23] and we confirm these results over a broader temperature range, see Fig. 2(a). We attribute these discrepancies to the presence of a finite fraction of localized modes. In fact, looking at the relation $e_s(f_{ud})$ in Fig. 2(a), where the temperature is used as an implicit variable, one observes a geometric transition at a finite energy threshold. The values of the threshold energies $e_{th}$, determined as the largest average energy of stationary points such that $f_{ud}(e_{th}) = 0$, are reported in Table 1. This behavior is confirmed in all of the models (except the network liquid, see below), and is a direct consequence of the localization transition.

To check the consistency between the analysis in terms of energy and temperature, we follow Refs. [9,11] and estimate the temperature at which the bare energy of the system, defined as $e_b(T) = e(T) - \frac{3}{2}T$ where $e(T)$ is the average potential energy per particle at temperature $T$, reaches the threshold energy. The idea is that the system in its thermal dynamics will sample stationary points through thermal activation when the average energy of the latter will be comparable to $e_b$, which provides an estimate of the energy of the potential well that confines the system. Figure. 2(b) shows that bare energy reaches $e_{th}$ very close to $T_{\mathrm{MCT}}$. Some deviations are seen for the polydisperse model with $n = 12$, consistent with the discrepancy seen in Fig. 1(b). The bare energy of this model reaches $e_{th}$ at $T_{th} = 0.080 \pm 0.009$, which is consistent with the localization transition temperature $T_\lambda = 0.086 \pm 0.002$ determined below. Overall, these findings suggest that $W$-minimizations provide a reasonable mapping to stationary points that are relevant to the thermal dynamics of the system.

One notable deviation from the pattern described above occurs for a model of a strong tetrahedral network liquid [37], which mimics the structure and dynamics of silica. Even though the vast majority of unstable modes of this model is localized at any temperature [37], a small fraction of delocalized modes survives even below the putative MCT crossover and no geometric transition is observed in the temperature range accessible to our simulations. While this discrepancy is in line with the conventional Angell picture, which asserts that fragile and

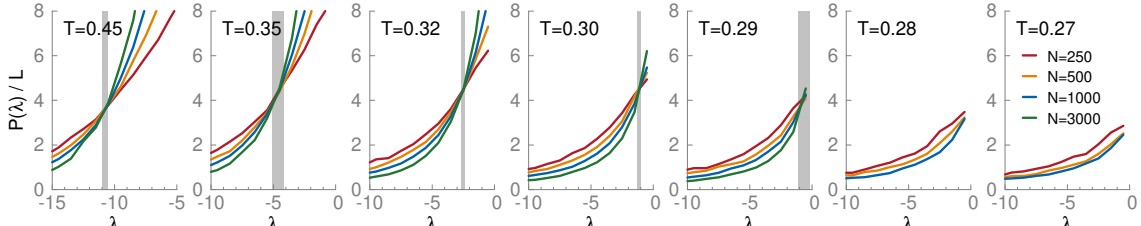

Figure 3: Scaled participation ratio $P(\lambda, N)/L$ of stationary and quasi-stationary points of the ternary mixture model from $W$-minimizations for all studied temperatures and system sizes $N$. The mobility edge $\lambda_e$ is the eigenvalue at which $P(\lambda, N)/L$ has a fixed point and is indicated by a vertical bar. The width of the vertical bar corresponds to the uncertainty on $\lambda_e$.

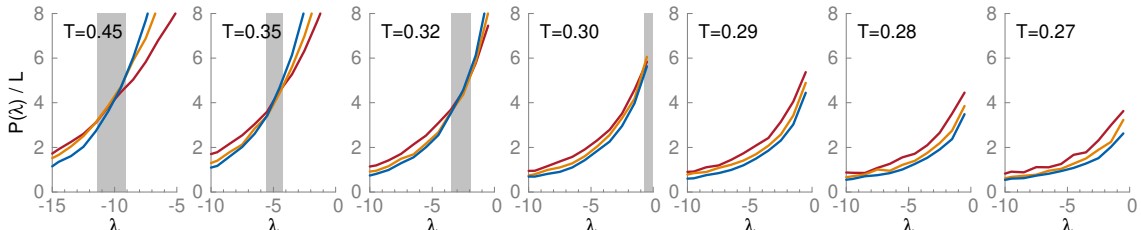

Figure 4: Scaled participation ratio $P(\lambda, N)/L$ of stationary points of the ternary model from EF optimizations for all studied temperatures and system sizes $N = 250, 500, 1000$. As in Fig. 3, the corresponding mobility edge is indicated by a vertical bar.

strong liquids form two distinct classes [1], it may also be due to the limited temperature range probed by our simulations. We also note that, since our optimization methods struggle to locate true stationary points for this model [1], these excess modes may be specific to quasi-stationary points. Nonetheless, we argue that even if an underlying geometric transition were present in this model, its features would be largely hidden by the large concentration of localized modes, which corresponds to the elementary rearrangements of the tetrahedral structural units [37]. At a more fundamental level, our results cast some doubts on the relevance of the MCT description of the early stages of glassy dynamics in strong liquids [43,44], see also [45].

The results presented above rely on our ability to determine the mobility edge of the spectrum, *i.e.* the eigenvalue $\lambda_e$ that separates localized and delocalized unstable modes. Early attempts to determine the mobility edge from the spectrum of the instantaneous normal modes (INM) in liquids were unsuccessfull [46], but the technical problems were recently tackled [47] by applying a finite-size scaling procedure borrowed from the study of Anderson localization [48]. We used this procedure to classify the unstable modes of all studied models. In the following, we illustrate this approach for one specific ternary mixture [35] and provide full details about the remaining systems in the SI and accompanying dataset [38].

As a measure of the localization of a mode $\alpha$, we consider the participation ratio

$$P_\alpha = \left( \sum_{i=1}^{N} |\vec{e}_{\alpha,i}|^4 \right)^{-1},$$

where $\vec{e}_{\alpha,i}$ is the displacement of particle $i$ along the corresponding normalized eigenvector. We then compute the average participation ratio $P(\lambda, N)$ of modes with eigenvalue $\lambda$ for a

---

[1]$W$-minimizations only located quasi-stationary points, while EF optimizations did not converge to stationary points within the prescribed number of 4000 iterations.

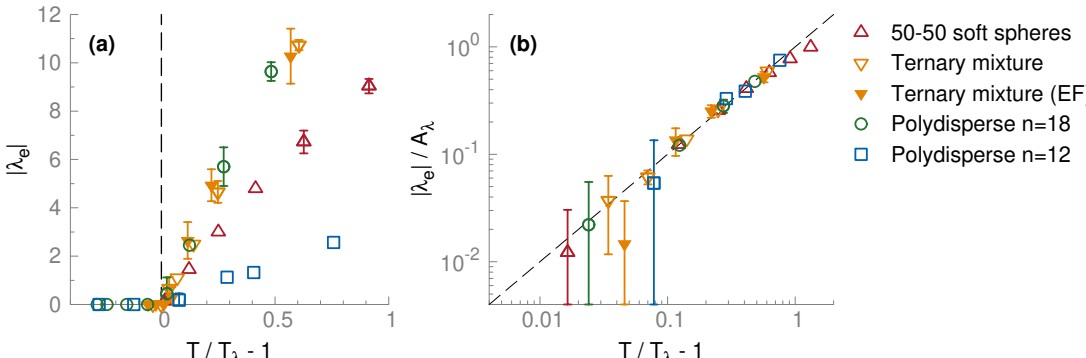

Figure 5: Mobility edge $|\lambda_e|$ as a function of rescaled temperature for all studied fragile liquids. Open and filled symbols correspond to $W$-minimizations an EF optimizations, respectively. The localization transition temperature $T_\lambda$ is extracted from a fit to Eq. (11). Panel (b) shows a log-log representation of $|\lambda_e|/A_\lambda$ versus the rescaled temperature. The dashed line represents the identity function.

given system size $N$. A finite-size scaling analysis allows one to determine the mobility edge $\lambda_e$ as the fixed point in $P(\lambda, N)/L$, where $L$ is the linear size of the system [47]. The rationale is that, for delocalized modes, $P$ scales at least linearly with $L$, whereas it is independent of $L$ for localized modes. As a result, the mobility edge identifies the eigenvalue where finite size effects change nature. Localized modes have $\lambda < \lambda_e$, while $\lambda_e < \lambda < 0$ for delocalized modes.

Figures 3 and 4 show $P(\lambda, N)/L$ across the MCT crossover temperature for the ternary mixture obtained from $W$-minimizations and EF optimizations, respectively. We use the ternary mixture as a bench case because it is the model for which we gathered the largest statistics. The other models display qualiatively similar features, see the SI [38]. A well-defined fixed point at $\lambda_e(T)$ is visible in the participation ratio when $T > T_{\text{MCT}}$. The data show that $\lambda_e$ reaches zero around $T = 0.28 \pm 0.01$, which matches well the mode-coupling temperature determined from the dynamics [27]. Below this temperature, the absence of a non-trivial fixed point means that all unstable modes are localized and we formally set $\lambda_e = 0$. Quantitatively, the mobility edge of the unstable modes was determined by finding the intersection of pairs of scaled participation ratios $P(\lambda, L_i)/L_i$ and $P(\lambda, L_j)/L_j$, where the indices $L_i$ and $L_j$ denote different linear system sizes. The mobility edge $\lambda_e$ is then defined as the average of the eigenvalues $\lambda_e^{ij}$ at which the scaled participation ratios cross. If two curves $P(\lambda, L_i)/L_i$ and $P(\lambda, L_j)/L_j$ do not intersect each other, the corresponding estimate $\lambda_e^{ij}$ of the mobility edge is set to zero. The uncertainty on $\lambda_e$ was estimated as half of the difference between the extreme values of $\lambda_e^{ij}$. To determine the uncertainty on the fractions of delocalized and localized unstable modes, we considered the $\lambda_e$-dependence of $f_{ud}(T; \lambda_e)$ and $f_{ul}(T; \lambda_e)$ respectively, and propagated the uncertainty on $\lambda_e$. Finally, we note that at low temperature, the unstable modes have a somewhat higher participation ratio in stationary points than in quasi-stationary points. However, the location of the mobility edge and its temperature dependence is unaffected by these slight discrepancies.

In Fig. 5 we show that the mobility edge goes to zero linearly as $T \to T_\lambda^+$ for all fragile liquids studied in this work and is zero for $T < T_\lambda$. The precise value of the localization temperature $T_\lambda$ can be extracted from least square fitting of the mobility edge to the following functional form

$$\lambda_e(T) = \begin{cases} A_\lambda(T/T_\lambda - 1) & \text{if } T > T_\lambda, \\ 0 & \text{if } T \leq T_\lambda \end{cases}, \tag{11}$$

where $A_\lambda$ and $T_\lambda$ are adjustable parameters. The values of $T_\lambda$ are reported in Table 1, they correlate strongly with those of $T_{\text{MCT}}$. The localization transition temperature $T_\lambda$, at which the

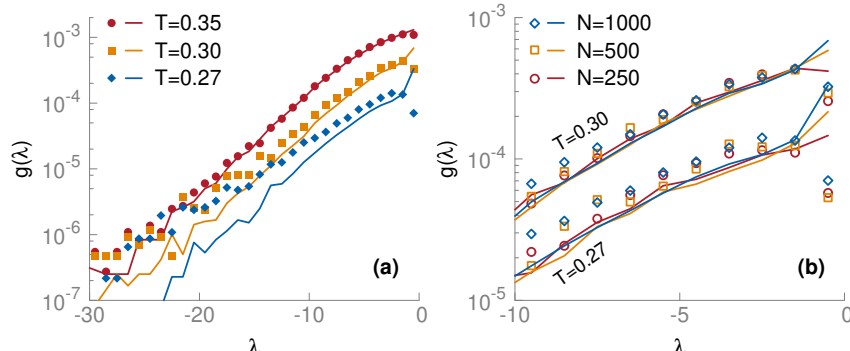

Figure 6: (a) Unstable part of the spectrum $g(\lambda)$ for the ternary mixture from $W$-minimizations (lines) and EF optimizations (symbols). (b) System size dependence of $g(\lambda)$ for stationary points obtained from EF optimizations.

mobility edge vanishes, defines a new characteristic temperature for glassy dynamics. It can be determined using a well-defined procedure and subsumes the conventional definition of the mode-coupling crossover based on fitting the dynamic data. Its quantitative determination is only limited by the statistical uncertainty on the mobility edge very close to the transition.

The fraction of delocalized and localized modes shown in Fig. 1 are then defined as $f_{ud} = \int_{\lambda_e}^{0^-} g(\lambda) d\lambda$ and $f_{ul} = \int_{-\infty}^{\lambda_e} g(\lambda) d\lambda$, respectively, where $g(\lambda)$ is the normalized spectrum of the stationary points [2]. We used the largest system size available for each system to compute the fractions of unstable modes, see the SI [38] for an analysis of finite size effects. Results obtained using INM, *i.e.* the normal modes of the instantaneous configurations, showed instead that the vast majority of the unstable directions are delocalized at any temperature [47], with no obvious change as $T \to T_{MCT}$. The INM thus seem unable to capture changes in the landscape properties relevant to the MCT crossover, unless perhaps by carefully filtering the unstable directions so as to remove inflection and non-diffusive modes [49]. These modes are not relevant for the dynamics and are suppressed by force minimizations and EF optimizations.

The shape of the unstable portion of the spectrum $g(\lambda)$ displays at all temperatures an exponential tail, as expected from an uncorrelated distribution of localized modes. This feature, which does not depend on the type of point, i.e. quasi-stationary or stationary point, is visible in Fig. 6(a) for the ternary mixture model and is confirmed for all studied models, see the SI [38]. As temperature decreases, however, the distributions of stationary points display one qualitative difference with respect to quasi-stationary points. As shown in Fig. 6(a), the spectrum of stationary points shows a depletion close to $\lambda = 0$. The depletion has a slight size dependence at low temperature, see Fig. 6(b). The suppression of these low frequency unstable modes has a counterpart in the temperature dependence of $f_{ud}$, which approaches zero more rapidly for stationary points than quasi-stationary points [see Fig. 1(b)]. By contrast, the spectrum of quasi-stationary points displays a slight excess of modes close to $\lambda = 0$, which gets more pronounced as $T$ decreases and $N$ increases. These features at small (absolute) eigenvalues are generic of all studied fragile liquids [38]. However, both types of points indicate a vanishing mobility edge close to $T_{MCT}$ and therefore these discrepancies do not affect the location of the localization transition. Our analysis demonstrates that the difference between stationary and quasi-stationary points does not boil down simply to the presence of an inflection mode, but reflects itself in the overall distribution of modes at small frequency $\omega = \sqrt{|\lambda|}$. In this regime, the shape of $g(|\omega|)$ suggests that the unstable saddle modes may share some

---

[2]The three null modes as well the as spurious inflection mode of quasi-stationary points [18, 19] are removed from the analysis.

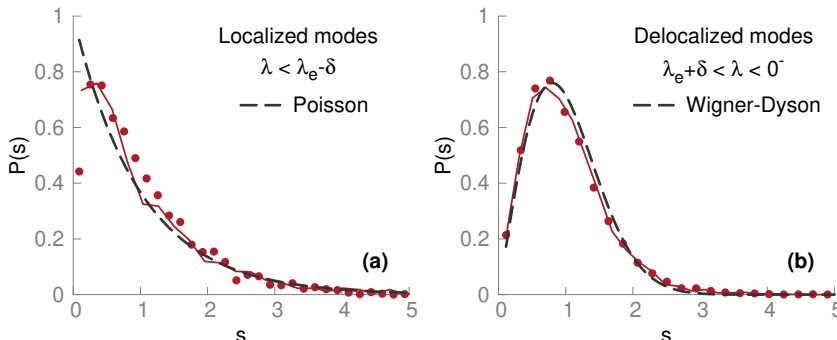

Figure 7: Level spacing statistics at $T = 0.35$ for (a) localized and (b) delocalized modes in the ternary mixture model. The distribution $P(s)$ of level spacings $s$ is computed in the indicated portion of eigenvalues, with $\delta = 2$. Full lines and symbols indicate results for $W$-minimizations ($N = 3000$) and EF optimizations ($N = 1000$), respectively. The dashed line indicates a Poisson distribution in (a) and a Wigner-Dyson distribution in (b), see text for definitions.

similarities with the quasi-localized stable modes [50], which arise from the hybridization of localized excitations and phonons [51] and display a $g(\omega) \sim \omega^4$ scaling [52, 53]. It will be interesting to analyze this connection in future work.

A well-established method to distinguish localized and delocalized modes builds on the analysis of the level spacing statistics [54], as used extensively in the analysis of spectra in a variety of problems, including the INM of supercooled and confined liquids [47, 55, 56]. Here we compute the level spacing distribution of stationary and quasi-stationary points of the PES. We first determine the smooth part of the cumulative density of states $\xi(\lambda)$ through a cubic spline of the raw data and then compute the level spacings $\tilde{s} = \xi(\lambda_{i+1}) - \xi(\lambda_i)$ from the ordered set of the eigenvalues. Finally, the level spacings are normalized, $s = \tilde{s}/\langle \tilde{s} \rangle$. In Fig. 7 we show representative results for the level spacing distribution at $T = 0.35$, i.e. in a regime where both kinds of modes can be clearly identified. We remove from the analysis the modes around the mobility edge (over a range $\pm 2$), for which the functional form of the level spacing statistics is known to have an intermediate character [54]. We find that the level spacing distribution of the delocalized modes $\lambda > \lambda_e$ is rather well described by the Wigner-Dyson distribution

$$P(s) = \left(\frac{\pi s}{2}\right) \exp\left(-\frac{\pi}{4} s^2\right).$$

For the localized modes, the distribution is close to the Poisson distribution expected for uncorrelated eigenvalues, $P(s) = \exp(-s)$, although there remains a small depletion at the smallest level spacings. The trend of the distributions for varying $N$ (not shown) suggests that this depletion may be a finite size effect. Overall, the analysis of the level spacing statistics corroborates our analysis of the localization of the unstable modes.

Finally, we make direct contact with the real space structure of the modes and inspect the average displacements on the unstable modes $e_i^2 = (1/n_u) \sum_\alpha |\vec{e}_{\alpha,i}|^2$. Since each eigenvector is normalized, we have $e_i^2 < 1$. By averaging over all unstable modes, the instability field $e_i^2$ captures the degree of localization of a given stationary point. The three snapshots in Fig. 8 depict the spatial distribution of $e_i^2$ for representative saddle points. On approaching $T_{\text{MCT}}$, the instability field becomes localized around few isolated particles with large displacements. It would be interesting to analyze separately the spatial correlation of localized and delocalized modes using for instance the gyration radius [37] or the methods of Ref. [57]. Early simulations [58] found a correlation between $e_i^2$ and the short-time dynamical heterogeneity of a Lennard-Jones mixture. Our results suggest that the growth of dynamic correlations associ-



Figure 8: Spatial distribution of the average displacements $e_i^2$ on the unstable modes for three stationary points sampled in the ternary mixture (a) well above ($T = 0.45$), (b) around ($T = 0.35$) and (c) at the mode-coupling temperature ($T = 0.29$). Only particles in a vertical slab of thickness $2\sigma$ are shown. The color coding interpolates between yellow (•) for particles with $e_i^2 = 0$ and red (•) for those with $e_i^2 \geq 0.04$.

ated to the progressive stabilization of unstable modes as $T \to T_{\mathrm{MCT}}$ is cut off because unstable modes become localized, which suggests a plausible explanation for the non-monotonic evolution of dynamic correlations across $T_{\mathrm{MCT}}$ observed in some liquids [34, 59–61]. Since localized modes are present at any temperature, see Fig. 1(c), supercooled liquids may display activated dynamics between nearest energy minima even above $T_{\mathrm{MCT}}$, at variance with the traditional Goldstein's scenario [8] but in agreement with studies on the metabasin structure of the landscape [21] and on dynamic facilitation [17].

## 5 Conclusions

In conclusion, we found that the localization properties of unstable directions of the potential energy landscape of several models of glasses display a qualitative change close to the mode-coupling crossover temperature $T_{\mathrm{MCT}}$. Our observations demonstrate that the geometric transition found in mean-field models and investigated in early simulation studies [9–11] involves only the subset of delocalized unstable modes. The mode-coupling crossover thus corresponds to a localization transition and is a meaningful physical concept only if the concentration of localized unstable modes is sufficiently low. These results may provide guidelines to understand the dynamic crossovers reported in some supercooled liquids by experiments [62, 63] and simulations [34, 59–61]. In liquids characterized by a high concentration of localized unstable modes, including *e.g.* strong glass-formers, the physics should be controlled instead by localized excitations, even above $T_{\mathrm{MCT}}$. Kinetically constrained models [64] could then provide an effective theoretical framework to account for the build up of dynamic correlations from such localized rearrangements. Liquids embedded in higher dimensions, which have recently received significant interest, are closer to mean-field behavior and we predict that they will display small concentrations of localized modes and be structurally very homogeneous. These expectations are consistent with recent findings for a nearly mean-field three-dimensional model [65], and can now be tested numerically in large dimensions [66]. Future studies should also focus on generalizing the analysis to models of experimentally relevant models of molecular liquids.

## Acknowledgements

We thank A. Ikeda, W. Kob, M. Ozawa, G. Pastore, M. Shimada and G. Tarjus for useful discussions. Post-processed data and workflow to reproduce the analysis and the figures of the article

and of the supplementary information is available at https://doi.org/10.5281/zenodo.1478600. This work was supported by a grant from the Simons Foundation (# 454933, L. Berthier).

# A    Quasi-stationary points versus stationary points

In this section we compare the statistical properties of quasi-stationary and stationary points obtained from $W$-minimizations. We focus mostly on the ternary mixture model, because it is the one for which we accumulated the largest statistics.

The plots in Fig. 9 shows results obtained separately for stationary points and for the bulk of the points obtained from $W$-minimizations for the ternary mixture model. Only minor discrepancies between the two sets of data are visible, the fraction of unstable modes being slightly smaller in stationary points at small $T$. No difference is visible in the geometric representation $f_u(e_s)$.

In Fig. 10 we show the participation ratio $P(\lambda)$ of the unstable modes for all points obtained from all $W$-minimizations and for stationary points only. The two sets of data are consistent with one another, with only minor discrepancies below the mode-coupling temperature $T_{\mathrm{MCT}} \approx 0.29$.

In Fig. 11(a) we show the fraction of unstable modes for stationary points only, *i.e.* without quasi-stationary points, for all fragile liquids and various system sizes. Within the noise of the data, we confirm that the fraction of unstable modes remains finite even below the mode-coupling crossover. See the Supplementary Information for further details on finite size effects.

Figure 11(b) is the same as Fig. 11(a) but for the fraction of delocalized unstable modes. We use the mobility edge obtained from analysis of all $W$-minimizations because the current statistics on the participation ratio is not sufficient to determine the mobility edge. The data are nicely consistent with Eq. (10). Close to $T_{\mathrm{MCT}}$, the fraction of delocalized unstable modes is lower in stationary points than in quasi-stationary points, consistent with the findings for EF optimizations. This feature can be traced back to the depletion in the spectrum of stationary points at small $\lambda$, see Fig. 6. Overall, the trend observed in Fig. 11 confirms the localization transition of the stationary points around the mode-coupling temperature.

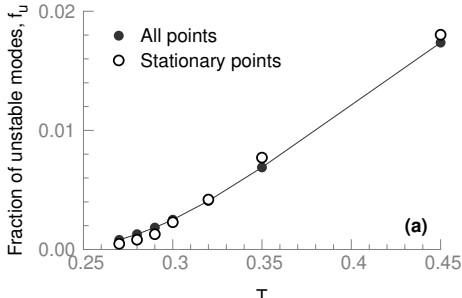
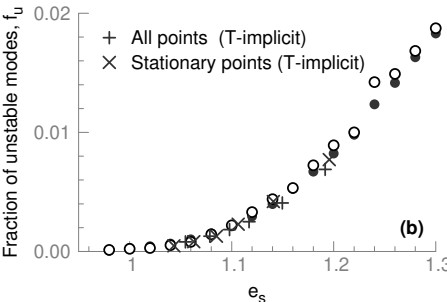

Figure 9: Fraction of unstable modes in all points and in true stationary points obtained from $W$-minimizations as function of (a) temperature and (b) energy. In panel (b) we show both averages on a per-energy basis (circles) and using $T$ as an implicit variable (crosses). The number of particles is $N = 500$.

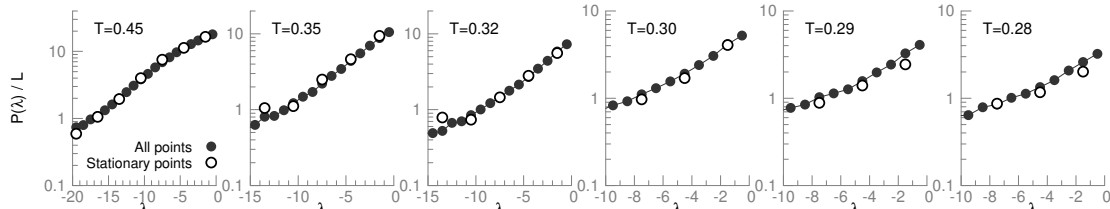

Figure 10: Scaled participation ratio $P(\lambda)/L$ as a function of $\lambda$ for the ternary mixture for all points obtained from $W$-minimizations (filled circles) and for stationary points only (empty circles). The number of particles is $N = 500$.

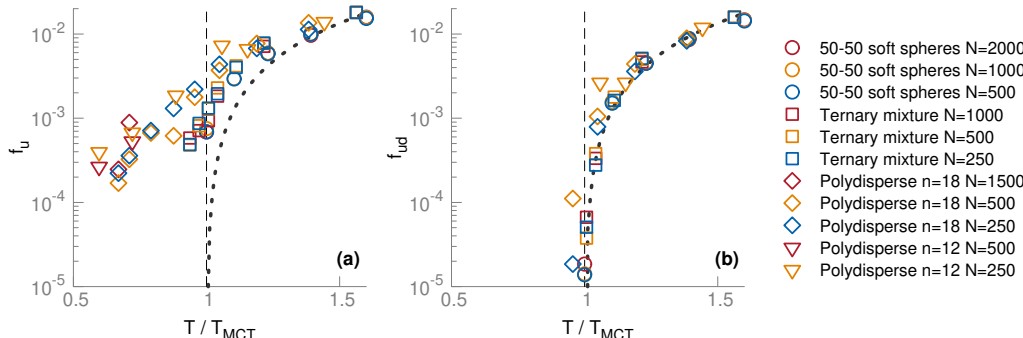

Figure 11: (a) Fraction of unstable modes $f_u$ as a function of $T/T_{\mathrm{MCT}}$ for stationary points obtained from $W$-minimizations in all studied models except the network-forming liquid. (b) Same as (a) but for the fraction of delocalized unstable modes $f_{ud}$. The correspodning system size $N$ is indicated in the legend. In both panels, the dotted line indicates the approximate master Eq. (10) for delocalized unstable modes as shown in Fig. 1(b). The vertical dashed lines indicates the location of the MCT crossover temperature.

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
