# Peer review of "A localization transition underlies the mode-coupling crossover of glasses"

_SciPost Physics, doi:SciPost Phys. 7, 077 (2019)_

## Round 2 · Referee Report · Anonymous (Referee 1) · 2019-6-6

Strengths

1-New approach to an old long standing debate (taking advantage of recent algorithmic and conceptual progresses)
2-Accurate analysis of the results under multiple perspectives

Weaknesses

1-The wording in the text is at times too terse, something could have been explained better.

Report

I am definitely in favour of publication after the following comments are taken into account and corresponding minor revisions are implemented in the text.

Requested changes

1-I do not see any reason to put Table 1 on second page. I found reference to that in the text only at pg.6, unless I am mistaken. It should be put in correspondence of the text referring to it or add a first introductory comment to the Table earlier on if felt important that it appears in first pages.

2-A few more words are to be devoted to explain what are quasi stationary points (it is not really clear as it is) and it should be clearly stated what it is the issue with them and how it is solved for the current data analysis (the current explanation is not well structure and a bit confusing).

3-The distribution of radii for the Polydisperse particles n=12 is not specified.

4-It is said that "the fraction of delocalized unstable modes [...] goes strictly to zero when the temperature drops below T_MCT". This seems an accurate description of data for 50-50 and ternary mixture, not quite so for the two polydisperse systems (n=12 and 18) and the network liquid for which the fraction of delocalized unstable modes goes to zero but somewhat below T_MCT. Nothing is said about this discrepancy. The Authors should comment more about that. Do they relate this to a small error in the estimation of T_MCT? Some finite size effects? Inflection points? Other reasons?

5-The sentence "Finally the trend in Fig. 1(c) superficially suggests a correlation between glass forming ability [27] and the concentration of localized unstable modes." must be expanded. In which way the suggested correlation goes? Can the Author spend a few more words on glass forming ability and what does this correlation, if any, imply?

6-When INM are mentioned on pg.7, it would be nice to read a brief explanation of why INM where not successful and in which way in comparison the adopted method is expected to be more reliable.

7-On pg.9 it would be beneficial to have some more words to introduce soft stable modes and to speculate about the proposed connection between their scaling with the one observed for the delocalized portion of the unstable spectrum.

8-Fig4 why data about the unstable part of the spectrum of stationary points are only presented for ternary mixtures? I did not find anywhere in text where it is commented if this result is robust across the different models studied. This should be clarified.

9-Why is the -average over all modes- used to find particles involved in the rearrangements? It is natural to ask whether different pictures emerge for modes with lambda>lambda_0 or lambda>lambda_0. May be a distribution of displacements for what is now labelled as a 'mobile' particle could show a non trivial (bimodal?) behaviour? I find the choice of focusing on the average a little coarse and simplistic.

  • validity: high
  • significance: high
  • originality: good
  • clarity: ok
  • formatting: good
  • grammar: good

Author:  Daniele Coslovich  on 2019-10-08  [id 625]

(in reply to Report 1 on 2019-06-06)

We thank the referee for his/her careful and constructive report. He/she provided a number of suggestions to improve the manuscript, which we mostly followed.

1-I do not see any reason to put Table 1 on second page. I found reference to that in the text only at pg.6, unless I am mistaken. It should be put in correspondence of the text referring to it or add a first introductory comment to the Table earlier on if felt important that it appears in first pages.

We agree with the referee, the table now appears in the Results section.

2-A few more words are to be devoted to explain what are quasi stationary points (it is not really clear as it is) and it should be clearly stated what it is the issue with them and how it is solved for the current data analysis (the current explanation is not well structure and a bit confusing).

Prompted by the remarks of both referees, we have signifincantly expanded our discussion and analysis of the similarities and differences between stationary and quasi-stationary points. We now include an additional set of optimizations obtained with the eigenvector-following method, see the Methods section as well as the reponse to the other referee for more details. These new data are presented and discussed along with the previous ones in the Results section. Additional comments on the distinction between quasi-stationary and stationary points have been added in the Introduction and in the Methods section.

3-The distribution of radii for the Polydisperse particles n=12 is not specified.

We have specified that the distribution is the same as for polydisperse particles with n=18.

4-It is said that "the fraction of delocalized unstable modes [...] goes strictly to zero when the temperature drops below T_MCT". This seems an accurate description of data for 50-50 and ternary mixture, not quite so for the two polydisperse systems (n=12 and 18) and the network liquid for which the fraction of delocalized unstable modes goes to zero but somewhat below T_MCT. Nothing is said about this discrepancy. The Authors should comment more about that. Do they relate this to a small error in the estimation of T_MCT? Some finite size effects? Inflection points? Other reasons?

We see two separate issues here. First, the fraction of delocalized modes for the network liquid does not go to zero within the explored range of temperature. This may be an intrinsic feature of this kind of liquids or a feature specific to quasi-stationary points. In fact, we could barely locate any stationary point for this model, no matter the optimization algorithm. We have added a related comment on this on page 8.

Concerning polydisperse particles, the fractions go to zero but at a temperature lower than T_MCT by about 10% (n=18) and 20% (n=12). These discrepancies are likely due to the well-known inherent uncertainty in fitting the MCT crossover from dynamic data. Or they might perhaps reflect a worse overall consistency with the geometric transition scenario. We have added a brief comment about this on page 7, pointing out the difficulties in determining the MCT crossover.

5-The sentence "Finally the trend in Fig. 1(c) superficially suggests a correlation between glass forming ability [27] and the concentration of localized unstable modes." must be expanded. In which way the suggested correlation goes? Can the Author spend a few more words on glass forming ability and what does this correlation, if any, imply?

We have expanded the corresponding discussion on page 7 and pointed to Ref. 27 for the discussion about the glass-forming ability of the studied models. We do not have a clear picture of what this correlation may imply and we prefer to leave this specific point open.

6-When INM are mentioned on pg.7, it would be nice to read a brief explanation of why INM where not successful and in which way in comparison the adopted method is expected to be more reliable.

To address this point, we have added a comment on page 10 about INM, pointing out which INM modes need to be filtered in order to establish contact with the MCT crossover, see Ref. 49.

7-On pg.9 it would be beneficial to have some more words to introduce soft stable modes and to speculate about the proposed connection between their scaling with the one observed for the delocalized portion of the unstable spectrum.

Prompted by the referee's remark, we have added a comment on page 11 about the quasi-localized modes observed in the stable, low frequency portion of the the spectrum. The possible connection with the unstable spectrum becomes even more intriguing in the light of the new data for stationary points, which shows a depletion close to lambda=0. Please see the new Fig. 6 and its corresponding discussion. Work to test this connection is currently under way.

8-Fig4 why data about the unstable part of the spectrum of stationary points are only presented for ternary mixtures? I did not find anywhere in text where it is commented if this result is robust across the different models studied. This should be clarified.

To address the point raised by the referee, we performed the following modifications:

  • We pointed out on page 11 which features are "universal" across models and/or type of points and which features are not.
  • We have added the full N- and T-dependence of the spectra in the Supplement section of the accompanying project document.

9-Why is the -average over all modes- used to find particles involved in the rearrangements? It is natural to ask whether different pictures emerge for modes with lambda>lambda_0 or lambda>lambda_0. May be a distribution of displacements for what is now labelled as a 'mobile' particle could show a non trivial (bimodal?) behaviour? I find the choice of focusing on the average a little coarse and simplistic.

We chose to show the average over all unstable modes because this quantity was found to correlate well with the propensity of motion in Ref. 58. We found that filtering the distribution P(e_u^i) according to the localization of the modes, as suggested by the referee, does not lead to interesting insight, even though his/her point is definitely valid. Promising alternative approaches to quantify the spatial correlations in the modes are the methods of Ref. 37 and Ref. 57, and we have added a related comment on page 12. Work in this direction is under way and will be presented in a separate publication.

---

## Round 2 · Referee Report · Anonymous (Referee 2) · 2019-6-14

Report

The manuscript revisits the issue of the geometrical transition in the potential energy landscape, which existence and possible link with the mode-coupling crossover of structural glasses was first discussed about 20 years ago. It is a welcome new exploration the subject, which now benefits from the existence of new models and increased computing power which allows to explore temperatures which were out of reach a decade ago.

The authors find saddles and "quasi-saddles" of the potential energy landscape and compute the fraction of unstable modes, $f_u$, and of unstable delocalized modes, $f_{ud}$ as a function of temperature for four model glass-forming liquids. The main message of the manuscript is this: from the numerical study of the four models, it is clear that nothing happens to the $f_u(T)$ curves near the mode-coupling temperature $T_\text{MCT}$. However, if one considers only delocalized unstable modes, one sees that the $f_{ud}(T)$ curves vanish near $T_\text{MCT}$ in a way compatible with a power-law singularity.

The subject and approach are interesting, and a welcome attempt to shed light on a difficult problem. Unfortunately, the manuscript has issues serious enough that I cannot recommend publication, at least in the present form.

  1. The main issue with this study is the very definition of the $f_u(T)$ and $f_{ud}(T)$ curves. While it is clear enough how to compute $f_u$ once a saddle point has been found, it is not clear how to associate a set of saddle points to a given temperature. In practice, one starts from an instantaneous equilibrium configuration and finds a saddle using some algorithm (in this case lBFGS minimization of the squared force), and associates the saddle to the temperature at which the instantaneous configuration was produced. However, in ref. 40 it was shown that this procedure leads to different $f_u(T)$ curves depending on the algorithm used to find saddle points. The problem is aggravated by the fact that with the algorithm used here, most of the time one does not find a saddle, but a "quasi-saddle" instead. Ref. 40 also showed that even when a saddle is found, it is not necessarily that which is closest to the instantaneous configuration. Although ref. 40 considered only $f_u(T)$, it is reasonable to presume that the same holds for $f_{ud}(T)$. Thus the curves of fig 1a and 1b are not very meaningful unless a strong case can be made for the use of one algorithm over another one. Of course it could be that different algorithms produce curves are different but vanish at the same temperature, but this remains to be shown. In any case, the exponent of eq. 9 is probaly meaningless.

  2. On the other hand, the $f(e)$ curves (where $e$ is the energy of the saddle) are much more robust (though ref. 40 found some small algorithm bias). But the comparison in fig 1d and 1e is not very meaningful: the threshold energy can be defined for $f_u(e)$ or for $f_{ud}(e)$ as the energy at which either of the two curves vanish. Since $f_{ud}(e)\le f_u(e)$ by definition, it is not surprising that $f_{ud}(e)$ vanishes at a higher energy than $f_u(e)$. The issue is which one of the two threshold energies can be linked to the $T_\text{MCT}$ crossover. For the reasons stated above, the $f(T)$ curves are suspect, so another approach is required. In references 9 and 11, it was argued that one should compare the energy of the saddles to the "bare" potential energy at equilibrium, defined as $u_\text{eq}(T)-3/2 k_B T$: the idea was to find the bottom of the potential energy well within which the system is moving. However, none of this is discussed in the manuscript.

  3. The significance of "quasi-saddles" is not discussed enough. While the present study confirms earlier results in the sense that the properties of the fraction of unstable modes behaves similarly in both saddles and "quasi-saddles", the dynamical relevance of the latter is not clear. It was argued in ref. 18 that "quasi-saddles" are not "quasi" at all, i.e. they are simply nonstationary points where the force is an eigenvector of the hessian with zero eigenvalue. While the fact that they behave similar to true stationary points is intriguing, the conceptual differences between the two call for an analysis of the $f(T)$ curves of true saddles in all the models considered.

  4. In p. 6 the authors write "These deviations were previously attributed to finitesize effects [40], but they actually stem from localized modes" (referring to deviations from linearity in the $f_u(e)$ curve near $e_\text{th}$. This statement is wrong: the comment in ref. 40 concerns the saddle point approximation, and applies both to $f_u(e)$ and $f_{ud}(e)$. If one defines the threshold energy as the point where the complexity $\Sigma(f,e)$ (where $f$ can be $f_u$ or $f_{ud}$) first has a maxium at $f=0$, then finite systems will actually have a small but nonzero average values of $f$ even below $e_\text{th}$ (because $f$ is a positive quantity, and saddle points continue to exist below the threshold, only that they become subdominant with respect to minima [or delocalized saddles]).

  5. Also in p. 6, "Such a representation [plotting $f$ vs. the energy of the saddle] reveals an intrinsic property of the landscape and does not depend on the way in which the latter is sampled". This is not quite true: ref. 40 found a small algorithm bias even in the $f(e)$ representation.

  • validity: low
  • significance: good
  • originality: good
  • clarity: high
  • formatting: good
  • grammar: excellent

Author:  Daniele Coslovich  on 2019-10-08  [id 624]

(in reply to Report 2 on 2019-06-14)

We thank the referee for his/her critical but constructive report, which prompted us to extend and improve our analysis. The referee raises some concerns about the minimization algorithm, but also suggests in point 2. a possible strategy to check the internal consistency of our results. We have thus addressed these concerns by (i) perfoming additional optimizations using a different method locating true saddle points and (ii) exploiting his/her suggestion in point 2. We also fully took into account the additional remarks raised in the report.

1) The main issue with this study is the very definition of the fu(T) and fud(T) curves. While it is clear enough how to compute fu once a saddle point has been found, it is not clear how to associate a set of saddle points to a given temperature. In practice, one starts from an instantaneous equilibrium configuration and finds a saddle using some algorithm (in this case lBFGS minimization of the squared force), and associates the saddle to the temperature at which the instantaneous configuration was produced. However, in ref. 40 it was shown that this procedure leads to different fu(T) curves depending on the algorithm used to find saddle points. The problem is aggravated by the fact that with the algorithm used here, most of the time one does not find a saddle, but a "quasi-saddle" instead. Ref. 40 also showed that even when a saddle is found, it is not necessarily that which is closest to the instantaneous configuration. Although ref. 40 considered only fu(T), it is reasonable to presume that the same holds for fud(T).

The main issue raised by the referee is not specific to our work and indeed has plagued previous research on this topic since the early studies of the Rome group in the early 2000's. The problem is quite intricate and somewhat technical, and we did our best to disentangle several sub-issues that the referee has condensed in the paragraph above. We also point out that a simple strategy to tackle the problem is offered by the referee himself/herself in point 2. and we address it below.

The first sub-issue concerns the algorithm-dependence of the curves f_u(T). The results presented in the previous version of the manuscript were based on force minimizations using the l-BFGS algorithm (W-minimizations). To address the referee's concerns, we considered the hybrid eigenvector-following (EF) method. In a first stage, we implemented the algorithm following Refs. 18, 20, but we encountered serious convergence issues. Similar issues were reported in the above papers (especially at high temperature) and can be alleviated by "fixing" the number of target unstable modes after a prescribed number, M, of iterations, as done already in Refs. 18, 20. This is the approach used in Ref. 32 (formerly Ref. 40) and which produced "algorithm-dependent" f_u(T) curves, as mentioned by the referee. We realized, however, that this dependence is inherent in the hybrid EF method: larger values of M leads to smaller values of f_u, but also decrease the success rate of the optimizations. On top of that, given M, the final value of f_u depends on the choice of the initial trust radius, which is a free parameter of the algorithm. Thus, the algorithm-dependence of the f_u(T) curves observed in Ref. 32 is partly artificial: the curves may be shifted upwards and downwards by choosing different values of M or of the initial trust radii. We also found a fundamental issue with the hybrid EF method: it displays an erratic behavior whenever the system passes through regions of zero curvature. This issue deteriorates the success rate of the optimizations (unless M is very small) and is not specific to the complex high-dimensional landscape of supercooled liquids: it can be reproduced under some circumstances even on the simple Muller-Brown surface. An account of these difficulties is given in the Methods section of the revised manuscript.

Based on these preliminary explorations and from a survey of the current literature (including a recent but unfortunately unpublished study by C. Ruscher, PhD thesis at the University of Strasbourg), we got to the unsatisfactory conclusion that the hybrid EF method does not provide a robust, parameter-free mapping between instantaneous configuration and stationary points. However, by fixing f_u from the outset, the original EF method can at least be used to address the second sub-issue mentioned by the referee: W-minimizations, while providing a well-defined mapping from initial configurations, only rarely locate true stationary points. Prompted by both referees, we decided to investigate this issue more carefully and we performed extensive searches of stationary points for the ternary mixture model using the EF method. Given the above problems, the value of f_u was fixed from the outset to the one of the point found by W-minimization starting from the same instantaneous configuration. This approach does not provide an independent mapping, of course, but it enables a straightforward and more extensive comparison between quasi-stationary and stationary points. Through this new analysis we evidenced a small qualitative difference between these two types of points which had so far passed undetected and which we further discuss below. The main point, after sorting out all these technical issues not mentioned in the referee's comment, is that the bulk of the conclusions at the core of the paper (localization transition of the unstable modes) remain unaffected. We include and discuss these new results throughout the revised manuscript.

The referee mentioned in passing a third sub-issue related to W-minimizations: Ref. 32 provides some evidence that the stationary points located by W-minimizations are not necessarily the "closest ones" to the corresponding instantaneous configurations. We performed additional analysis and qualitatively confirmed this behavior, however we also observed that: (i) the difference between the distances of the W-minimized point and the closest stationary point is on average rather small (of the order of 10%, consistent with Ref. 32) and crucially, (ii) the average fraction f_u of the closest points is statistically indistinguishable from the one of the bulk of the points. Thus, even though W-minimizations do not always locate the closest points, the mapping that defines f_u(T) is unaffected and remains rather well-defined.

Thus the curves of fig 1a and 1b are not very meaningful unless a strong case can be made for the use of one algorithm over another one. Of course it could be that different algorithms produce curves are different but vanish at the same temperature, but this remains to be shown. In any case, the exponent of eq. 9 is probaly meaningless.

Our take on this is that there exists, to date, no satisfactory mapping between instantaneous configurations and stationary points for high-dimensional glassy landscapes (methods like Newton homotopy still have to prove their relevance for this field). Nonetheless, the conclusion we got after 3 months of exploration of this intricate problem is that, despite the known issue about quasi-stationary points, W-minimizations provide a rather well-defined mapping, i.e. f_u(T), which is more robust than the one of the alternative hybrid EF. Some evidence that this mapping is not irrelevant to the dynamics was given long ago in Ref. 58.

We may agree that the agreement with the exponent found in mean-field p-spin models is a bit surprising (rather than "meaningless"), but it is also clear from the text that we do not make a strong case for it: we only state that the data can be "described approximately" by the power law in Eq. 10 with exponent 3/2. Actually, the agreement with the new set of stationary points is quite good.

Finally, a simple strategy to circumvent the above issues comes from a suggestion of the referee in point 2, to which we now arrive.

2) On the other hand, the f(e) curves (where e is the energy of the saddle) are much more robust (though ref. 40 found some small algorithm bias). But the comparison in fig 1d and 1e is not very meaningful: the threshold energy can be defined for fu(e) or for fud(e) as the energy at which either of the two curves vanish. Since fud(e)≤fu(e) by definition, it is not surprising that fud(e) vanishes at a higher energy than fu(e). The issue is which one of the two threshold energies can be linked to the TMCT crossover. For the reasons stated above, the f(T) curves are suspect, so another approach is required. In references 9 and 11, it was argued that one should compare the energy of the saddles to the "bare" potential energy at equilibrium, defined as ueq(T)−3/2kBT: the idea was to find the bottom of the potential energy well within which the system is moving. However, none of this is discussed in the manuscript.

We thank the referee for this suggestion. We determined the temperature at which the bare energy, which is a proxy to the inherent structure energy, crosses the threshold energy determined from the vanishing of delocalized modes f_ud. We found that it matches well the MCT crossover temperatures for all models except the polydisperse n=12. A similar discrepancy is observed in Fig. 1(b) and is likely due to the inherent uncertainties and ambiguities in fitting the dynamic data (choice of the temperature range). Indeed, the temperature at which the crossing occurs agrees well with the localization temperature T_lambda. This new analysis is included in the new Fig.2, which now gathers all the plots on the energy-dependence of the data. The fact that the bare energy reaches the threshold energy around the localization transition is a nice confirmation of our previous analysis. It also corroborates the validity of the mapping f_u(T) determined from W-minimizations, since it provides quantitatively similar localization temperatures, thus solving the intricate "algorithm-dependence" issue mentioned in point 1. above.

We would also like to emphasize that, while it is clear that f_ud(e_s) must vanish at a T higher or equal than f_u(e_s), our central result is that f_ud(e) vanishes for all fragile liquids, while f_u(e_s) does not (not even in the thermodynamic limit, see point 4. below), except perhaps for the 50-50 mixture. This finding is another strong indication that f_ud is much more relevant than f_u to establish the connection with the MCT crossover. We trust that these results will fully address the referee's concerns.

3) The significance of "quasi-saddles" is not discussed enough. While the present study confirms earlier results in the sense that the properties of the fraction of unstable modes behaves similarly in both saddles and "quasi-saddles", the dynamical relevance of the latter is not clear. It was argued in ref. 18 that "quasi-saddles" are not "quasi" at all, i.e. they are simply nonstationary points where the force is an eigenvector of the hessian with zero eigenvalue. While the fact that they behave similar to true stationary points is intriguing, the conceptual differences between the two call for an analysis of the f(T) curves of true saddles in all the models considered.

Prompted by both referees, we have addressed this issue in greater detail. Our main conclusions on this point are:

  • We found small qualitative differences in the unstable portion of the spectrum g(lambda) at small eigenvalues. Stationary points sampled at energies around and below the threshold energy show a depletion of unstable modes close to zero, whereas quasi-stationary points have a slight excess. These features came to our eyes by zooming in the low-frequency portion of the spectrum with better resolution and statistics. We thus agree with the referee that quasi-stationary points sampled at low energy are not just stationary points with an extra inflection mode. The finite force associated to these points reflect itself in a more subtle way in the distribution of the unstable modes. At higher energy, the spectra of both types of points are (for a fixed value of f_u) essentially indistinguishable.
  • The above differences appear only around and below the threshold energy, in a regime where the unstable modes are localized. We found that the unstable modes found at low energy and small eigenvalues have slightly higher participation ratios in stationary points found by EF than quasi-stationary points. The location of the mobility edge is, however, unaffected and it still vanishes around T_MCT.

Throughout the revised manuscript we discuss in much more detail the similarities and differences between the two types of points, now that we have excellent statistics for both. In particular, we point out on page 11 that "the difference between stationary and quasi-stationary points does not boil down simply to the presence of an inflection mode" but also that these discrepancies do not affect the localization properties.

Prompted by the referee's final remark, we have also added a panel to Fig.11 in the Appendix showing the curves f_u(T) for true stationary points for all models and system sizes for which these points could be sampled. The comparison of the available data for f_u and f_ud for true stationary points support our conclusions about the unstable modes localization.

4) In page 6 the authors write "These deviations were previously attributed to finite size effects [40], but they actually stem from localized modes" (referring to deviations from linearity in the fu(e) curve near eth. This statement is wrong: the comment in ref. 40 concerns the saddle point approximation, and applies both to fu(e) and fud(e). If one defines the threshold energy as the point where the complexity Σ(f,e) (where f can be fu or fud) first has a maximum at f=0, then finite systems will actually have a small but nonzero average values of f even below eth (because f is a positive quantity, and saddle points continue to exist below the threshold, only that they become subdominant with respect to minima [or delocalized saddles]).

We removed that specific sentence about finite size effects from page 7. However please note the following:

  • Delocalized unstable modes are defined in this work using a strict threshold on the eigenvalue, which removes finite size effects almost by construction. Therefore, the argument of Ref. 32 cannot be applied to delocalized modes. An alternative approach would be to set a threshold on the participation ratio and analyze the system size dependence of the resulting fraction of delocalized modes, along the lines of Ref. 32.
  • The full f_u(e_s) does not show appreciable finite size effects in the models that could be equilibrated below the MCT crossover. These finite size effects would appear if saddles became subdominant wrt minima. We have added figures in the Supplementary Information to show that there are essentially no finite size effects on the f_u(e_s) curves in the localized regime.
  • We note that the argument of Ref. 32 may perhaps apply to the 50-50 soft spheres, but it is difficult to gather equilibrium data at low temperature for this model because of its tendency to crystallize.

5) Also in page 6, "Such a representation [plotting f vs. the energy of the saddle] reveals an intrinsic property of the landscape and does not depend on the way in which the latter is sampled". This is not quite true: ref. 40 found a small algorithm bias even in the f(e) representation.

We have revised this sentence, which now reads "an intrinsic property of the landscape and it is fairly insensitive to the way in which the latter is sampled."

---

## Round 3 · Author Response

We thank the referees for their competent and constructive reports. We have addressed their criticisms and concerns as follows:

  • We have finally performed additional, extensive searches for 'true' stationary points using a different optimization method (eigenvector-following). The new data are presented and discussed in the revised version and allow us to investigate closely the similarities and differences between stationary and quasi-stationary points, as requested by both referees.

  • Prompted by referee 2, we have performed additional analysis of the energy-dependence of stationary points (as opposed to their temperature-dependence). We found that the bare energy reaches the threshold energy at a temperature close to the MCT crossover, in agreement with our analysis in terms of temperature. This new analysis is presented in the new Fig. 2 and corroborates the mapping between instantaneous configurations and stationary points provided by force minimizations, thereby addressing the main concern of referee 2.

  • We have substantially expanded and improved the discussion in the text, to make the presentation more accessible and to clarify all the unclear points indicated by the referees.

  • We also carefully checked again the numerics of all our optimizations, enforcing strict convergence criteria for all studied models.

See our replies to the referees' reports for full details.

With these extensive set of changes, we feel that the quality of the manuscript has substantially improved and that the main physical message of our work (the localization of unstable modes around the MCT crossover) has passed a fairly stringest test. We hope therefore that it will be considered suitable for publication in SciPost.

---

## Round 3 · List of Changes

• Add eigenvector-following optimizations for the ternary mixture model. The figures and accompanying discussions have been updated to reflect this new set of data.
  • Add analysis of the bare energy as a function of temperature, as suggested by referee 2.
  • New Fig. 2, which gathers the analysis on the energy-dependence of the fractions of modes.
  • Extend and revise the presentation of the unstable spectrum g(lambda), see new Fig. 6.
  • Add panel to the new Fig. 11 in the Appendix, showing f_u(T) for stationary points in all studied models.
  • Add plots of the unstable spectrum of all studied models in the Supplementary information.
  • Add plots of N-dependence of f_u(e_s) of all studied models in the Supplementary information.
  • Several minor fixes and improvements to the text to take into account the remarks of the referees.

---

## Editorial Decision

published